# Efficient Reasoning via Reward Model

## Abstract

Reinforcement learning with verifiable rewards (RLVR) has been shown to enhance the reasoning capabilities of large language models (LLMs), enabling the development of large reasoning models (LRMs). However, LRMs such as DeepSeek-R1 and OpenAI o1 often generate verbose responses containing redundant or irrelevant reasoning step—a phenomenon known as overthinking—which substantially increases computational costs. Prior efforts to mitigate this issue commonly incorporate length penalties into the reward function, but we find they frequently suffer from two critical issues: length collapse and training collapse, resulting in sub-optimal performance. To address them, we propose a pipeline for training a Conciseness Reward Model (CRM) that scores the conciseness of reasoning path. Additionally, we introduce a novel reward formulation named Conciseness Reward Function (CRF) with explicit dependency between the outcome reward and conciseness score, thereby fostering both more effective and more efficient reasoning. From a theoretical standpoint, we demonstrate the superiority of the new reward from the perspective of variance reduction and improved convergence properties. Besides, on the practical side, extensive experiments on five mathematical benchmark datasets demonstrate the method's effectiveness and token efficiency, which achieves an 8.1% accuracy improvement and a 19.9% reduction in response token length on Qwen2.5-7B. Furthermore, the method generalizes well to other LLMs including Llama and Mistral. The implementation code and datasets are publicly available for reproduction: `https://anonymous.4open.science/r/CRM`.

## 1 Introduction

Recent advances in large reasoning models (LRMs) like DeepSeek-R1 (Guo et al., 2025) validate that reinforcement learning with verifiable reward (RLVR) methods like GRPO (Shao et al., 2024) could bring more powerful reasoning capabilities. However, it was found that the LRMs like o1 (OpenAI, 2024) tend to generate verbose and redundant responses with repetitive and irrelevant steps (Luo et al., 2025) (like unnecessary verification at $x = 0$ on model after GRPO training, as shown in Sec. 4.6). This '**over-thinking**' (Chen et al., 2024) issue introduces significantly heavy computational overhead.

Consequently, efficient reasoning (Sui et al., 2025) is introduced to seek for both concise reasoning path and strong reasoning capabilities. Among the existing works (Sui et al., 2025), a common strategy is to introduce a length-related reward to penalize long reasoning path and combine with the original verifiable outcome reward through addition operation like in Kimi 1.5 (Team et al., 2025), cosine scale reward (Yeo et al., 2025), ALP (Xiang et al., 2025), and LC-R1 (Cheng et al., 2025).

Nevertheless, we find two critical challenges of these methods on efficient reasoning. First, as shown in the training curves of GRPO on Qwen2.5-7B using cosine scale reward in Fig. 1(a), a counterintuitive phenomenon emerges: *the generated token length drastically declines while the reward still increases steadily.* Referring to the case study in Sec. 4.6,

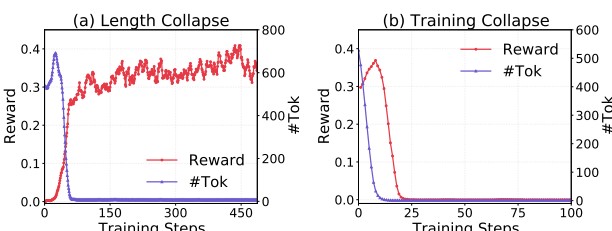

Figure 1: (a) Length collapse and (b) Training collapse phenomenon. '#Tok' denotes the number of tokens generated.

we can see that the LLM's capacity for logical reasoning has been impaired, reducing it to a state of rote memorization of the final answer. We refer to this phenomenon as '**length collapse**', which

indicates reward hacking (Skalse et al., 2022). Second, as shown in the training curves of GRPO on Qwen2.5-7B using ALP (Xiang et al., 2025) in Fig. 1(b), *both the reward and number of tokens drastically decline*. This phenomenon is known as '**training collapse**'. Notably, we also observe these two phenomena on other LLM backbones and existing methods, which indicates the limitations of simply evaluating the conciseness of reasoning path by token length of in-batch samples.

To address them, we propose a framework to achieve efficient reasoning via reward model, which consists of the following two steps. 1) A pipeline of training a conciseness reward model is proposed. Specifically, Qwen2.5-Math-72B-Instruct and Qwen2.5-72B-Instruct are first adopted for data augmentation and labeling tasks on public mathematical data, respectively. Then supervised fine-tuning is conducted to train the reward model. Consequently, given a mathematical question and the corresponding reasoning path (i.e., solution), this reward model is capable of evaluating the conciseness of solution and generating a conciseness score. 2) In RL training, we design conciseness reward function consisting of explicit dependency between outcome reward and conciseness score to mitigate reward hacking. We also justify the benefit of this new design in optimization through theoretical analysis. Besides, an annealing and difficulty-related coefficient are introduced to stabilize training and adapt response length to estimated question difficulty, respectively.

Our key contributions are summarized as follows:

- To the best of our knowledge, it is the first work to identify and address the length collapse and training collapse problem in existing works on tackling over-thinking of large reasoning models.
- We propose a pipeline to train a conciseness reward model (CRM) which is able to evaluate the conciseness of reasoning path from multiple aspects and generate a conciseness score.
- We design conciseness reward function (CRF) with explicit dependency between the outcome reward and conciseness score of reasoning path, i.e., the conciseness score is applied only when the answer is correct. This dependency yields two advantages: variance reduction and improved convergence rate in optimization. Notably, the theoretical proof of them is also provided.
- Extensive experiments on five representative mathematical benchmark datasets show that, first, on Qwen2.5-7B the proposed framework achieves an improvement of 8.1% in accuracy and a 19.9% reduction in token length compared with the original GRPO using binary outcome reward. Besides, it also shows effectiveness and compatibility with different LLM backbones like Llama and Mistral.

## 2 PRELIMINARY

In this section, we provide the background of Group Relative Policy Optimization (GRPO) (Shao et al., 2024), which is a representative reinforcement learning algorithm in reinforcement learning with verifiable rewards (RLVR). Specifically, the LLM with parameters $\theta$ is denoted as a policy model $\pi_\theta$. In each iteration, the LLM takes a given query $q$ (e.g., a mathematical question) as input and samples a group of $G$ independent responses $\{o_i\}_{i=1}^G$ consisting of tokens $\{o_{i,t}\}_{t=1}^{|o_i|}$ from the old policy model $\pi_{\theta_{\text{old}}}$. Then it is optimized by maximizing the following simplified objective:

$$\mathcal{J}_{\text{GRPO}}(\theta) = \mathbb{E}_{q \sim P(Q),\, \{o_i\}_{i=1}^G \sim \pi_{\theta_{\text{old}}}(\cdot|q)} \frac{1}{G} \sum_{i=1}^G \frac{1}{|o_i|} \sum_{t=1}^{|o_i|}$$

$$\left\{ \min \left[ \frac{\pi_\theta(o_{i,t} \mid q, o_{i,<t})}{\pi_{\theta_{\text{old}}}(o_{i,t} \mid q, o_{i,<t})} A_{i,t}, \text{clip}\left( \frac{\pi_\theta(o_{i,t} \mid q, o_{i,<t})}{\pi_{\theta_{\text{old}}}(o_{i,t} \mid q, o_{i,<t})}, 1-\varepsilon, 1+\varepsilon \right) A_{i,t} \right] - \beta \mathbb{D}_{\text{KL}}\left(\pi_\theta \| \pi_{\text{ref}}\right) \right\} \tag{1}$$

where $\varepsilon$ is a clipping-related hyper-parameter to stabilize training and $A_{i,t}$ denotes the advantage. Specifically, $A_{i,t} = \frac{R_i - \mu_R}{\sigma_R + \varepsilon'}$ where $\mu_R$, $\sigma_R$, and $\varepsilon'$ are the mean of rewards $\{R_i\}_{i=1}^G$, standard deviation of $\{R_i\}_{i=1}^G$, and a small constant. The reward function $R$ is usually set as binary verifiable reward considering the correctness of answer:

$$R_i^o = \begin{cases} 1 & \text{if the answer of } o_i \text{ is correct} \\ 0 & \text{otherwise} \end{cases}$$

The coefficient $\beta$ of the KL-divergence penalty is set to 0 in this paper for simplicity in optimization, which follows the same setting of the recent works (Yu et al., 2025; Yan et al., 2025; Liu et al.,

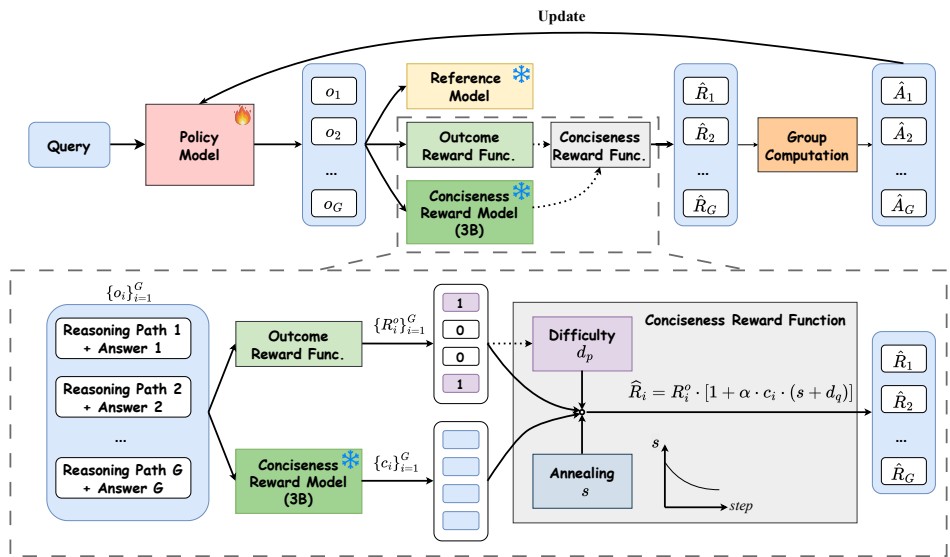

Figure 2: An overview of the proposed framework taking GRPO as an example.

2025b). Nonetheless, as introduced in Sec. 1, the model trained by GRPO algorithm using this pure outcome-related reward tend to result in over-thinking phenomenon.

## 3 METHOD

### 3.1 OVERVIEW

The overview of our proposed framework is depicted in Fig. 2 taking GRPO as an example, which consists of two steps. First, a conciseness reward model is trained to score on the conciseness of the reasoning path. Second, in the RL training, we design a new reward function to combine the outcome reward and the conciseness score generated from the trained conciseness reward model. In the following, the two steps are detailed in Sec. 3.2 and 3.3, then we provide two propositions to justify the advantage of conciseness reward function and discuss the proposed method.

### 3.2 CONCISENESS REWARD MODEL

To align with human preferences (Christiano et al., 2017; Ouyang et al., 2022), the most common way is to train a reward model from the data annotated by human in reinforcement learning from human feedback (RLHF). However, acquiring such high-quality human preference data is exceptionally resource-intensive, which necessitates recruiting large cohorts of human annotators to rank, compare, or rate model-generated responses. Besides, it also faces the challenges from a potential lack of domain-specific expertise, the absence of a unified scoring principle, as well as inconsistent preferences among individuals.

Therefore, to tackle these challenges, we propose a pipeline to train a conciseness reward model (CRM) which is capable of evaluating and generating score on the conciseness of reasoning path. Specifically, the pipeline consists of the following four steps.

- To begin with, we select DeepMath-103K (He et al., 2025) dataset and implement data augmentation to obtain the data for reward model training because: 1) It contains a substantially higher proportion of problems with higher difficulty. 2) It includes distinct reasoning paths generated from DeepSeek-R1 (Guo et al., 2025). 3) It has broad diversity covering various mathematical topics. 4) It undergoes decontamination procedure.

- Next, two distinct versions of solution are generated by Qwen2.5-Math-72B-Instruct model: one characterized by greater conciseness and the other by increased redundancy given the mathematical question and the original DeepSeek-R1 solution. Afterward, Qwen2.5-72B-Instruct model is employed to score on the conciseness of reasoning path ranging from 0.1 to 1 considering different

factors including avoiding repetition, eliminating irrelevant steps, and minimizing token length while maintaining solution quality. The prompt design is detailed in Appendix. A.

- Furthermore, to ensure the discriminability of the data, only redundant solutions with lower conciseness scores and concise solutions with higher conciseness scores are retained.

- Finally, we obtain an enriched high-quality dataset with 65878 samples and it is split into the training and validation set. The training set is leveraged to train the conciseness reward model through supervised fine-tuning (SFT) and Qwen2.5-3B-Instruct is adopted as the backbone model. We also provide an example of scoring on conciseness in Appendix. B.

### 3.3 CONCISENESS REWARD FUNCTION

After obtaining the conciseness reward model (CRM), it is able to evaluate and provide a score $c_i$ on the conciseness of the reasoning path of the generated responses $o_i$ to diverse mathematical questions in different topics. However, as introduced in Sec. 1, simply using the weighted sum of the outcome reward and conciseness score would probably result in length collapse and training collapse.

Therefore, our core motivation and novelty lies in investigating a better way to combine the outcome reward and conciseness score of reasoning path. Specifically, we design a new reward function named conciseness reward function (CRF) with explicit dependency between the outcome reward and conciseness score, i.e., the conciseness score for the reasoning path is applied only when the answer is correct to alleviate reward hacking. To justify the advantage of such design over the common weighted sum operation, on the one hand, we provide two propositions in the following Sec. 3.4. On the other hand, the experimental results of ablation study are shown in Sec. 4.4 comparing the full model with the model variant using weighted sum (denoted as 'w/o Dep').

Furthermore, to stabilize training, an annealing coefficient $s = \exp(-\frac{step}{T})$ is introduced that decays with increasing training steps where $step$ and $T$ denote the current training step and the total number of training steps, respectively.

Besides, on the DeepMath-103K dataset we observe that questions with higher difficulty levels tend to have longer solutions. For example, solutions to questions of difficulty 6.5–7 are 15.0% longer than those of difficulty 4–4.5 on average. Accordingly, we argue that it is justifiable to impose a reduced length penalty coefficient on harder questions, and conversely, a larger one on easier questions. Therefore, we evaluate the difficulty of the current question for the policy model based on the accuracy of the group answer (Fan et al., 2025; Xiang et al., 2025; Liu et al., 2025b), and calculate a difficulty coefficient $d$ for each question $q$ accordingly:

$$d_q = \exp(\frac{|\{i|R_i^o = 1, i = 1 \cdots G\}|}{G})$$

where $|\cdot|$ denotes the cardinality of a set and this coefficient ranges from 1 to e.

Finally, our proposed reward function CRF can be formulated as:

$$\hat{R}_i = R_i^o \cdot [1 + \alpha \cdot c_i \cdot (s + d_q)]. \tag{2}$$

When it is introduced into GRPO (Shao et al., 2024), the optimization objective would be:

$$\mathcal{J}_{\text{GRPO}}^{\text{CRF}}(\theta) = \mathbb{E}_{q \sim P(Q), \{o_i\}_{i=1}^G \sim \pi_{\theta_{\text{old}}}(\cdot|q)} \frac{1}{G} \sum_{i=1}^G \frac{1}{|o_i|} \sum_{t=1}^{|o_i|}$$

$$\left\{ \min \left[ \frac{\pi_\theta(o_{i,t} \mid q, o_{i,<t})}{\pi_{\theta_{\text{old}}}(o_{i,t} \mid q, o_{i,<t})} \hat{A}_{i,t}, \text{clip} \left( \frac{\pi_\theta(o_{i,t} \mid q, o_{i,<t})}{\pi_{\theta_{\text{old}}}(o_{i,t} \mid q, o_{i,<t})}, 1 - \varepsilon, 1 + \varepsilon \right) \hat{A}_{i,t} \right] - \beta \mathbb{D}_{\text{KL}} \left( \pi_\theta \| \pi_{\text{ref}} \right) \right\} \tag{3}$$

where it differs from the original objective $\mathcal{J}_{\text{GRPO}}$ only in the advantage that $\hat{A}_{i,t} = \frac{\hat{R}_i - \mu_{\hat{R}}}{\sigma_{\hat{R}} + \varepsilon'}$ where $\mu_{\hat{R}}$ and $\sigma_{\hat{R}}$ denote the mean and standard deviation of $\{\hat{R}_i\}_{i=1}^G$, respectively.

Besides, it can also be easily applied in different state-of-the-art RL algorithms like DAPO (Yu et al., 2025). To sum up, CRF possesses the following characteristics: 1) Only providing positive reward for correct answer. 2) Providing higher reward for more concise solution evaluated by conciseness reward model. 3) Providing lower coefficient for harder questions. 4) Providing lower weight of conciseness score with increasing training steps.

## 3.4 DISCUSSIONS

In this section, first the benefits of the proposed reward function are justified, which originate from the explicit dependency between the outcome reward and the conciseness score. This dependency leads to the following two propositions:

> **Proposition 1. Variance Reduction**: *Under the assumption that conciseness reward $c_i$ is positively correlated with outcome reward $R_i^o$ where $Cov(R_i^o, c_i) > 0$, the modified GRPO objective that incorporates a conciseness reward $J(\theta)$ exhibits reduced variance of stochastic gradient estimator $\nabla_\theta J(\theta)$ compared with original GRPO objective $J_o(\theta)$:*
>
> $$Var[\nabla_\theta J(\theta)] \leq (1 - \eta) Var[\nabla_\theta J_0(\theta)]$$
>
> *for some $\eta > 0$ that depends on the strength of the correlation between $R_i^o$ and $c_i$.*

The proof of Proposition 1 is detailed in Appendix C.1. The implementation of a conciseness reward is crucial for variance reduction in the GRPO framework. By acting as a dense signal, the conciseness reward addresses the issue of sparse and binary outcome rewards, which typically lead to high-variance gradient estimates. This reduction in variance makes each policy update more reliable. Instead of the policy jumping around erratically due to noisy estimates, the model takes more consistent steps in the right direction.

Given this proven variance reduction property, we further hypothesize that the modified reward will lead to a better convergence rate for our policy optimization algorithm.

> **Proposition 2. Improved Convergence Constants**: *Consider the stochastic gradient ascent update for the GRPO framework:*
>
> $$\theta_{t+1} = \theta_t + \alpha_t \hat{g}_t$$
>
> *where $\hat{g}_t$ is an unbiased estimator of the gradient $\nabla_\theta J(\theta_t)$, satisfying the variance bound $\mathbb{E}[\|\hat{g}_t - \nabla_\theta J(\theta_t)\|^2] \leq \sigma_g^2$. Assume the policy function $\pi_\theta(o|q)$ is twice continuously differentiable with respect to $\theta$, and there exist constants $L_1, L_2 > 0$ such that $\|\nabla_\theta \pi_\theta(o|q)\| \leq L_1$ and $\|\nabla_\theta^2 \pi_\theta(o|q)\| \leq L_2$. For a specific choice of step sizes $\alpha_t = \frac{\alpha}{\sqrt{t}}$ where $\alpha > 0$, the modified objective function $J(\theta)$, achieves a convergence rate of $O(1/\sqrt{T})$ with improved constants:*
>
> $$\mathbb{E}\left[\frac{1}{T} \sum_{t=1}^{T} \|\nabla_\theta J(\theta_t)\|^2\right] \leq \frac{2(J(\theta^*) - J(\theta_1)) + \alpha^2 G^2 + \alpha^2 \sigma_g^2}{\alpha \sqrt{T}} \leq \mathbb{E}\left[\frac{1}{T} \sum_{t=1}^{T} \|\nabla_\theta J_o(\theta_t)\|^2\right]$$
>
> *where $T$ is the total number of training steps, $\theta^*$ is a stationary point.*

The formal proof of Proposition 2 is detailed in Appendix C.2.

Meanwhile, beyond facilitating more powerful reasoning ability in mathematics and more concise reasoning path, our proposed framework, which consists of reward model construction and post-training with CRF, is adaptable to different human-specified objectives.

## 4 EXPERIMENTS

In the experiment part, we try to answer the following research questions:

- **RQ1:** How does our framework perform compared with state-of-the-art baseline methods?

- **RQ2:** Is CRF compatible with different LLM backbones?

- **RQ3:** What are the effects of different components in the new reward function?

- **RQ4:** What is the impact of hyper-parameter $\alpha$ combining outcome reward and conciseness score?

- **RQ5:** What are the differences in the reasoning path generated by our method compared to other baseline methods?

### 4.1 EXPERIMENTAL SETTINGS

In the following, we introduce the datasets, evaluation metrics, and baseline methods. The implementation details of experiments are provided in Appendix. D.

#### 4.1.1 DATASETS

As mentioned in Sec. 3.2, DeepMath-103K (He et al., 2025) dataset is processed for reward model training and RL post-training in which the train and validation set contain 31296 and 1643 samples, respectively. Besides, five mathematical benchmarks datasets including MATH-500 (Hendrycks et al., 2021), AIME2025, OlympiadBench (He et al., 2024), AMC-23, and GPQA-Diamond (Rein et al., 2024) are selected for evaluation.

#### 4.1.2 EVALUATION METRICS

To conduct evaluation, the Pass@$k$ metric is adopted, which measures the proportion of questions for which at least one of the $k$ generated solutions is correct. Besides, the number of tokens of solution is used to evaluate token efficiency. Therefore, a higher Pass@$k$ and lower number of the generated response tokens indicate better performance.

#### 4.1.3 BASELINES

We compare our method with the following representative baseline methods:

- **SFT** represents conducting supervised fine-tuning on the training set. Specifically, the mathematical question is regarded as query while the reasoning path together with the answer is seen as response.
- **ARM** (Wu et al., 2025) proposes trained with Ada-GRPO with format diversity reward to support adaptive reasoning formats. For fair comparison, we evaluate its performance of the public model checkpoint in the short CoT mode using Qwen2.5-7B as the backbone model.
- **Cos** (Yeo et al., 2025) denotes the cosine scale reward, which is proposed to stabilize the length scaling of RL training and control CoT length. Its reward function can be formulated as $R_i = R_i^o + \alpha \cdot f(R_i^o, L_i)$ where $R_i^o$ denotes the correctness, $\alpha$ is the hyper-parameter, $L_i$ is the generation length of $o_i$, and

$$f(R_i^o, L_i) = \begin{cases} \text{CosFn}(L_i, L_{\max}, r_0^c, r_L^c) & \text{if } R_i^o = 1 \\ \text{CosFn}(L_i, L_{\max}, r_0^w, r_L^w) & \text{if } R_i^o = 0 \\ r_e & \text{if } L_i = L_{\max} \end{cases}$$

where $\text{CosFn}(t, T, \eta_{\min}, \eta_{\max}) = \eta_{\min} + \frac{1}{2}(\eta_{\max} - \eta_{\min})(1 + \cos(\frac{t\pi}{T}))$. $L_{\max}$, $r_0^c$, $r_L^c$, $r_0^w$, $r_L^w$, and $r_e$ are hyper-parameters.

- **Kimi** (Team et al., 2025) represents the Kimi 1.5 reward which adds a length reward to the original outcome reward to improve token efficiency. Its reward function can be formulated as $R_i = R_i^o + \alpha \cdot f(R_i^o, L_i)$ where $R_i^o$ denotes the correctness, $\alpha$ is the hyper-parameter, $L_i$ is the generation length of $o_i$, and

$$f(R_i^o, L_i) = \begin{cases} \lambda & \text{if } R_i^o = 1 \\ \min(0, \lambda) & \text{if } R_i^o = 0 \end{cases}$$

$$\lambda = 0.5 - \frac{L_i - \min_{i=1}^G L_i}{\max_{i=1}^G L_i - \min_{i=1}^G L_i}$$

### 4.2 OVERALL PERFORMANCE (RQ1)

To answer RQ1, we choose Qwen2.5-7B as the LLM backbone and directly conduct RL training (same as Zero-R1 (Liu et al., 2025a)). The training curves are depicted in Fig. 3(a). Specifically, we compare the performance of CRF with different baseline methods introduced in Sec. 4.1.3 and the overall performance is shown in Tab. 1. We have the following observations.

First, compared to SFT, GRPO achieves a 14.2% improvement in accuracy but at the cost of a 43.9% increase in token length on average. Second, though cosine reward and kimi 1.5 reward significantly

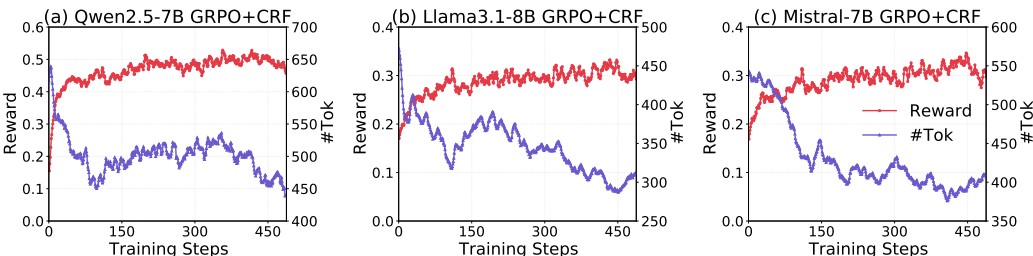

Figure 3: Training curves of outcome reward and average number of tokens of the generated reasoning path on (a) Qwen2.5-7B, (b) Llama3.1-8B, and (c) Mistral-7B-v0.1

Table 1: Overall performance comparison on MATH-500, AIME2025, OlympiadBench, AMC-23, and GPQA with Qwen2.5-7B. We report Pass@1 accuracy (%) and the average token length of response (#Tok). Bold denotes the best result.

| Datasets | MATH-500 | | AIME2025 | | OlympiadBench | | AMC-23 | | GPQA | | Average | |
| Metrics | Pass@1 | #Tok | Pass@1 | #Tok | Pass@1 | #Tok | Pass@1 | #Tok | Pass@1 | #Tok | Pass@1 | #Tok |
|---|---|---|---|---|---|---|---|---|---|---|---|---|
| SFT | 68.2 | 408.3 | 0.0 | 588.2 | 34.4 | 553.8 | 35.0 | 590.1 | 34.3 | 447.9 | 34.4 | 517.7 |
| ARM | 73.6 | 649.5 | **10.0** | 2649.4 | 41.3 | 1318.8 | 50.0 | 1329.3 | 33.3 | 575.3 | 41.6 | 1304.5 |
| GRPO | **78.2** | 535.5 | 6.7 | 992.5 | **39.2** | 776.5 | 42.5 | 794.0 | 29.8 | 627.5 | 39.3 | 745.2 |
| +Cos | 63.4 | 291.4 | 0.0 | 806.4 | 34.0 | 519.2 | 37.5 | 391.4 | 33.8 | 308.3 | 33.7 | 463.3 |
| +Kimi | 66.4 | 302.1 | 0.0 | 669.9 | 33.2 | 515.9 | 47.5 | 541.1 | 32.3 | 88.4 | 35.9 | 423.5 |
| +CRF | 76.0 | 398.2 | 6.7 | 916.7 | 37.7 | 649.1 | **55.0** | 648.9 | **36.9** | 373.3 | **42.5** | 597.2 |

reduce the token length by 37.8% and 43.2%, the accuracy decrease by 14.2% and 8.7%, respectively. Third, even if the short CoT mode of ARM is selected, it achieves a 5.9% improvement in accuracy but brings a significant improvement of 75.1% in the number of tokens generated compared to GRPO. By contrast, our method obtains a 8.1% improvement in accuracy by and a 19.9% reduction in token length compared to GRPO, indicating a better trade-off between effectiveness and efficiency.

## 4.3 COMPATIBILITY (RQ2)

Apart from Qwen2.5-7B, we also experiment on representative LLMs including Llama3.1-8B and Mistral-7B-v0.1. However, we observe the cold-start problem, i.e., directly training the base model with RL (same as Zero-R1) yields significantly worse performance than supervised fine-tuning. Further illustrations are provided in Appendix. E. Therefore, for these two backbones the R1-like training is adopted, i.e., we first conduct SFT on the base model then implement RL-based post-training. The results are shown in Tab. 2.

Similar to Qwen2.5-7B, we can see that the cosine scaled reward and Kimi 1.5 reward can significantly shorten the reasoning path but at the cost of diminished accuracy in solving mathematical problems. By contrast, compared to the original GRPO, our method achieves a 3.1% increase in accuracy and a 39.0% decrease in the response token length on Llama3.1-8B. Besides, it obtains a 9.1% increase in and a 17.7% decrease in the response token length on Mistral-7B-v0.1. Consequently, our method is justified to be effective and compatible with different LLM backbones.

## 4.4 ABLATION STUDY (RQ3)

To investigate the effects of each component in the proposed conciseness reward, we conduct ablation study on Qwen2.5-7B and the evaluation results on the validation set are depicted in Fig. 4(a). Specifically, we compare the full model with the model variant without annealing (denoted as w/o Ann), without the dependency between outcome reward and conciseness score (denoted as w/o Dep), without difficulty (denoted as w/o Dif), and without both

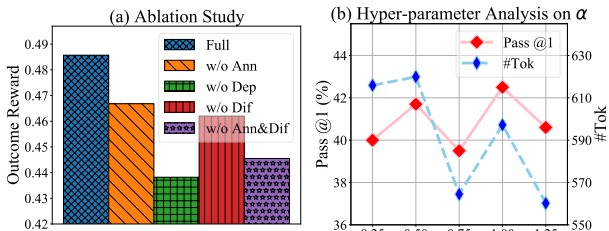

Figure 4: (a) Ablation study where the y-axis denotes the outcome reward on the validation set. (b) Hyper-parameter analysis on $\alpha$ where the left and right y-axis denote Pass@1 and number of tokens.

Table 2: Overall performance comparison on MATH-500, AIME2025, OlympiadBench, AMC-23, and GPQA with Llama3.1-8B and Mistral-7B-v0.1. We report Pass@1 accuracy (%) and the average token length of response (#Tok). Bold denotes the best result.

| Llama3.1-8B | MATH-500 | | AIME2025 | | OlympiadBench | | AMC-23 | | GPQA | | Average | |
| --- | --- | --- | --- | --- | --- | --- | --- | --- | --- | --- | --- | --- |
| | Pass@1 | #Tok | Pass@1 | #Tok | Pass@1 | #Tok | Pass@1 | #Tok | Pass@1 | #Tok | Pass@1 | #Tok |
| SFT | **29.8** | 516.4 | **3.3** | 950.3 | 5.6 | 646.9 | 5.0 | 553.4 | 27.8 | 498.4 | 14.3 | 633.1 |
| GRPO | 28.8 | 598.6 | 0.0 | 876.1 | **9.5** | 777.4 | 10.0 | 683.5 | 31.3 | 549.2 | 15.9 | 697.0 |
| +Cos | 10.0 | 10.5 | 0.0 | 8.1 | 4.3 | 10.8 | 7.5 | 8.9 | 25.8 | 7.3 | 9.5 | 9.1 |
| +Kimi | 21.4 | 97.0 | 0.0 | 189.9 | 6.8 | 87.2 | 10.0 | 74.6 | 28.3 | 38.8 | 13.3 | 97.5 |
| +CRF | 29.6 | 340.1 | 0.0 | 701.1 | 8.2 | 414.0 | **12.5** | 344.5 | **31.8** | 326.4 | **16.4** | 425.2 |
| Mistral-7B | MATH-500 | | AIME2025 | | OlympiadBench | | AMC-23 | | GPQA | | Average | |
| | Pass@1 | #Tok | Pass@1 | #Tok | Pass@1 | #Tok | Pass@1 | #Tok | Pass@1 | #Tok | Pass@1 | #Tok |
| SFT | 19.0 | 564.9 | 0.0 | 728.9 | 4.9 | 664.0 | 2.5 | 531.5 | 29.3 | 504.7 | 11.1 | 598.8 |
| GRPO | 15.2 | 575.3 | 0.0 | 656.6 | 4.9 | 617.0 | **7.5** | 548.1 | 27.3 | 565.9 | 11.0 | 592.6 |
| +Cos | 8.8 | 16.8 | 0.0 | 7.6 | 5.6 | 8.6 | 5.0 | 7.7 | 29.3 | 7.0 | 9.7 | 9.5 |
| +Kimi | 10.0 | 8.9 | 0.0 | 8.0 | 4.6 | 9.6 | 5.0 | 7.7 | **29.8** | 7.0 | 9.9 | 8.2 |
| +CRF | **20.0** | 396.6 | 0.0 | 735.8 | **5.9** | 468.2 | 5.0 | 441.6 | 29.3 | 395.1 | **12.0** | 487.5 |

annealing and difficulty (denoted as w/o Ann&Dif). First and foremost, when the explicit dependency between outcome reward and conciseness score is replaced with common weighted sum operation, the outcome reward drops by 9.8%. Besides, compared to the full model, the performance decreases by 3.9%, 4.9%, and 8.3% on w/o Ann, w/o Dif, and w/o Ann&Dif , respectively, which validates the effectiveness of the proposed annealing strategy and difficulty-related design.

### 4.5 HYPER-PARAMETER ANALYSIS (RQ4)

Taking Qwen2.5-7B as an example, we search for the optimal $\alpha$ from 0.25 to 1.25. The results are shown in Fig. 4(b) and we found $\alpha = 1$ achieves the best accuracy with relatively short token length. Similarly, we also search for $\alpha$ on Llama3.1-8B and Mistral-7B-v0.1 and the selected value is 0.5.

### 4.6 CASE STUDY (RQ5)

We conduct a case study on a simple math question and compare the output of model trained by GRPO, GRPO with cosine reward (denoted as GRPO+Cos), GRPO with Kimi 1.5 reward (denoted as GRPO+Kimi), and GRPO with our conciseness reward function (denoted as GRPO+CRF). The results are shown below. It can be observed that, first, all methods answer correctly but GRPO generates verbose and redundant reasoning path. Second, GRPO+Cos undergoes reward hacking with rote memorization of the final answer. Finally, our method generates shorter reasoning path than GRPO+Kimi, indicating better trade-off between accuracy and token efficiency while maintaining the logical quality of reasoning path.

`Question`: Find all solutions to the inequality $\frac{x^2}{(x-3)^2} \geq 0$. (Give your answer in interval notation.) Let's think step by step and output the final answer within \boxed{}.

---

*GRPO:* To solve the inequality $\frac{x^2}{(x-3)^2} \geq 0$, we need to analyze the expression and determine where it is non-negative.\n\n1. **Identify the critical points:**\n - The numerator is $x^2$, which is zero when $x = 0$.\n - The denominator is $(x-3)^2$, which is zero when $x = 3$.\n - The expression is undefined at $x = 3$ because division by zero is not allowed.\n\n2. **Analyze the sign of the expression:**\n - For $x \neq 3$, both $x^2$ and $(x-3)^2$ are always non-negative (since they are squares of real numbers).\n - Therefore, the fraction $\frac{x^2}{(x-3)^2}$ will be non-negative for all $x \neq 3$.\n - At $x = 0$, the expression equals 0, which satisfies the inequality $\frac{x^2}{(x-3)^2} \geq 0$.\n\n3. **Consider the interval notation:**\n - The expression is defined and non-negative for all $x$ except $x = 3$.\n - Including $x = 0$, the solution set is all real numbers except $x = 3$.\n\nThus, the solution in interval notation is $(-\infty, 3) \cup (3, \infty)$.\n\nThe final answer is \boxed{$(-\infty, 3) \cup (3, \infty)$}.

---

> *GRPO+Cos:* \boxed{$(-\infty, 3) \cup (3, \infty)$}

> *GRPO+Kimi:* To solve the inequality $\frac{x^2}{(x-3)^2} \geq 0$, we note that the numerator $x^2$ is always non-negative, and the denominator $(x-3)^2$ is always positive except when $x = 3$, where it is zero. Therefore, the expression is non-negative for all $x$ except $x = 3$. \boxed{$(-\infty, 3) \cup (3, \infty)$}

> *GRPO+CRF:* The inequality $\frac{x^2}{(x-3)^2} \geq 0$ holds when the numerator is non-negative and the denominator is non-zero. Since $x^2 \geq 0$ for all real $x$ and $(x-3)^2 > 0$ for all $x \neq 3$, the inequality is satisfied for all $x \neq 3$. Therefore, the solution is:\boxed{$(-\infty, 3) \cup (3, \infty)$}

## 5 RELATED WORK

In this section, we summarize the related works on reinforcement learning with verifiable rewards (RLVR) and methods tackling overthinking of large reasoning models.

### 5.1 REINFORCEMENT LEARNING WITH VERIFIABLE REWARDS (RLVR)

The notion of reinforcement learning with verifiable rewards (RLVR) originates from DeepSeek-Math (Shao et al., 2024) and Tulu 3 (Lambert et al., 2024). Specifically, it employs verifiable reward functions that evaluate the correctness of model outputs via deterministic rules, yielding a binary reward signal. Notably, its integration dramatically enhances accuracy in math reasoning, coding, and different domains (Su et al., 2025).

However, a key limitation of the existing RLVR methods is the inability of the rule-based reward to scale to complex tasks. Besides, either manually designed reward function or reward signal generated only from reward model tend to be vulnerable to reward hacking (Skalse et al., 2022). By contrast, we design a new reward function with sequential dependency between rewards, and provide theoretical proof on its better properties of variance reduction and convergence rate.

### 5.2 OVERTHINKING OF REASONING

Although RLVR empowers LLM with improved reasoning ability, it is observed that large reasoning models that undergo RL as the post-training step tend to over-think, i.e., generating verbose thinking steps. To tackle this, a common strategy is penalizing long responses through different reward function designs (Team et al., 2025; Yeo et al., 2025; Xiang et al., 2025; Cheng et al., 2025). For example, ALP (Xiang et al., 2025) dynamically adjust the weight of the length penalty based on the difficulty of the question. LC-R1 (Cheng et al., 2025) adopts both length reward and compress reward to remove the invalid part in reasoning path. However, as analyzed in Sec. 2, these methods are suspect to length collapse and training collapse, thus achieving sub-optimal performance. By contrast, our solution with the trained conciseness reward model and the proposed new reward function succeeds in addressing these issues.

## 6 CONCLUSION

In this paper, we first identify the length collapse and training collapse issues in the existing works on mitigating overthinking of large reasoning models post-trained by reinforcement learning. Therefore, to achieve more accurate and more efficient reasoning, we propose to train a conciseness reward model (CRM) to score on the conciseness of reasoning path and introduce a new reward function CRF with explicit dependency between the outcome reward and the conciseness score. On the one hand, compared with common weighted sum of reward, we justify the superiority of our reward function through theoretical analysis on its superior properties, i.e., variance reduction and improved convergence. On the other hand, extensive experiments on five mathematical benchmark datasets demonstrate the effectiveness and efficient reasoning of the proposed method, which also shows compatibility with different LLM backbones.

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

## A  PROMPT TEMPLATE

Taking Qwen2.5-7B as an example, the prompt template of evaluating the conciseness of thinking process is provided in Fig. 5.

## B  EXAMPLE OF CONCISENESS SCORE

In the following, we provide an example of a mathematical question, two reasoning paths of its solutions, and the corresponding conciseness score.

**Prompt Template**

```
<|im_start|>system
```
# Mathematical Solution Conciseness Evaluation\nEvaluate the conciseness of the mathematical solution based on three key factors: avoiding repetition, eliminating irrelevant steps, and minimizing token length while maintaining solution quality.\n## Task\nGiven mathematical question and its solution, score the solution's conciseness from 0.1-1, where higher scores indicate better efficiency in reasoning presentation.\n## Conciseness Evaluation Factors\n### 1. Repetition Avoidance\n- **High Score**: No repeated calculations, explanations, or reasoning patterns\n- **Low Score**: Multiple instances of similar steps, redundant explanations, additional confirmation or verification, or repeated mathematical operations\n### 2. Step Relevance\n- **High Score**: Every step directly contributes to solving the problem\n- **Low Score**: Includes tool use like python code, tangential explanations, unnecessary background, or steps that don't advance the solution\n### 3. Token Efficiency\n- **High Score**: Achieves complete solution with minimal word count and optimal mathematical notation\n- **Low Score**: Verbose explanations, excessive words where symbols suffice, unnecessarily long descriptions\n## Scoring Framework\n**0.9-1 (Highly Concise)**:\n- Zero repetition of steps or explanations\n- All steps are directly relevant to the solution\n- Minimal token usage with maximum information density\n- Optimal use of mathematical notation over verbose text\n**0.7-0.8 (Concise)**:\n- Very minimal repetition\n- Nearly all steps are relevant with minor exceptions\n- Efficient token usage with good information density\n**0.5-0.6 (Moderately Concise)**:\n- Some repetitive elements or slightly irrelevant steps\n- Moderate token efficiency - could be shortened without losing clarity\n**0.3-0.4 (Somewhat Verbose)**:\n- Notable repetition or several irrelevant steps\n- Higher token count than necessary for the solution complexity\n**0.1-0.2 (Very Verbose)**:\n- Significant repetition and irrelevant content\n- Excessive token usage with poor information-to-length ratio\n- Extra confirmation or verification using tools like python code\n## Response Format\nOnly provide a single integer (0.1-1) for the solution. No additional text or explanation. Be strict, accurate, and objective.
```
<|im_end|>
<|im_start|>user
```
Question: {Question}
Solution: {Solution}
```
<|im_end|>
<|im_start|>assistant
```

Figure 5: Prompt template used to evaluate conciseness of thinking process.

`Question:` Given the equations $25(9x^2 + y^2) + 9z^2 - 15(5xy + yz + 3zx) = 0$ and $x + y + z = 18$, find all possible even integer values of $y$ that are less than 15. Let's think step by step and output the final answer within `\boxed{}`.

**Conciseness Score: 0.3** To solve the given problem, we start with the equations $25(9x^2 + y^2) + 9z^2 - 15(5xy + yz + 3zx) = 0$ and $x + y + z = 18$. We need to find all possible even integer values of $y$ that are less than 15.\n\nFirst, we substitute $z = 18 - y - x$ into the first equation:\n\n$25(9x^2 + y^2) + 9(18 - y - x)^2 - 15(5xy + y(18 - y - x) + 3x(18 - y - x)) = 0$.\n\nExpanding and simplifying, we get:\n\n$25(9x^2 + y^2) + 9(324 - 36y - 36x + y^2 + 2xy + x^2) - 15(5xy + 18y - y^2 - yx + 54x - 3xy - 3x^2 - 3xy) = 0$, \n$225x^2 + 25y^2 + 2916 - 324y - 324x + 9y^2 + 18xy + 9x^2 - 15(5xy + 18y - y^2 - yx + 54x - 3xy - 3x^2 - 3xy) = 0$,\n$234x^2 + 34y^2 + 2916 - 324y - 324x + 18xy - 15(5xy + 18y - y^2 - yx + 54x - 3xy - 3x^2 - 3xy) = 0$,\n$234x^2 + 34y^2 + 2916 - 324y - 324x + 18xy - 15(5xy + 18y - y^2 - yx + 54x - 3xy - 3x^2 - 3xy) = 0$,\n$234x^2 + 34y^2 + 2916 - 324y - 324x + 18xy - 15(5xy + 18y - y^2 - yx + 54x - 3xy - 3x^2 - 3xy) = 0$, **(Many repetitions)** \n$279x^2 + 49y^2 + 3xy - 1134x$

$-594y+2916 = 0.$\n\nWe treat this as a quadratic in $x$:\n\n$279x^2+(3y-1134)x+(49y^2-594y+2916) = 0.$\n\nFor $x$ to be an integer, the discriminant of this quadratic equation must be a perfect square. The discriminant $D$ is given by:\n\n$D = (3y-1134)^2 - 4 \cdot 279 \cdot (49y^2 - 594y + 2916).$\n\nSimplifying the discriminant, we find:\n\n$D = -54,675y^2 + 656,100y - 1,968,300.$\n\nThis discriminant must be non-negative for real solutions. Solving the inequality $-54,675y^2 + 656,100y - 1,968,300 \geq 0$, we simplify it to:\n\n$(y-6)^2 \leq 0.$\n\nThis implies $y = 6$ is the only solution. Substituting $y = 6$ into the original equations confirms that $x = 2$ and $z = 10$ satisfy both equations.\n\nThus, the only even integer value of $y$ less than 15 is \boxed{6}.

**Conciseness Score: 0.8** Given the equations $25(9x^2 + y^2) + 9z^2 - 15(5xy + yz + 3zx) = 0$ and $x + y + z = 18$, we need to find all possible even integer values of $y$ that are less than 15.\n\nFirst, we substitute $z = 18 - y - x$ from the second equation into the first equation. After expanding and simplifying, we obtain the quadratic equation in $x$ and $y$:\n\n$279x^2 + 49y^2 + 3xy - 1134x - 594y + 2916 = 0$\n\nWe treat this as a quadratic in $x$:\n\n$279x^2 + (3y - 1134)x + (49y^2 - 594y + 2916) = 0$\n\nFor $x$ to be an integer, the discriminant of this quadratic equation must be a perfect square. The discriminant $D$ is given by:\n\n$D = (3y - 1134)^2 - 4 \cdot 279 \cdot (49y^2 - 594y + 2916)$\n\nSimplifying the discriminant, we find:\n\n$D = -54,675y^2 + 656,100y - 1,968,300$\n\nThis discriminant must be non-negative for real solutions. Solving the inequality $-54,675y^2 + 656,100y - 1,968,300 \geq 0$, we simplify it to:\n\n$(y - 6)^2 \leq 0$\n\nThis implies $y = 6$ is the only solution. Substituting $y = 6$ into the original equations confirms that $x = 2$ and $z = 10$ satisfy both equations. \n\nThus, the only even integer value of $y$ less than 15 is \boxed{6}.

## C  THEORETICAL PROOF

### C.1  THE PROOF OF PROPOSITION 1

Given $\Delta_a(\theta)$ is the correction term due to conciseness, the gradient variance of stochastic gradient estimator $\nabla_\theta J(\theta)$ can be decomposed as:

$$\text{Var}[\nabla_\theta J(\theta)] = \text{Var}[\nabla_\theta J_0(\theta)] + \text{Var}[\Delta_a(\theta)] + 2\text{Cov}[\nabla_\theta J_0(\theta), \Delta_a(\theta)] \tag{4}$$

The correction term variance is bounded by:

$$\text{Var}[\Delta_a(\theta)] \leq \frac{a^2 C_1}{N} \tag{5}$$

for some constant $C_1$ that depends on the variance of conciseness and gradient norms.

Leverage positive covariance when $\text{Cov}(R_i^o, c_i) > 0$:

$$\text{Cov}[\nabla_\theta J_0(\theta), \Delta_a(\theta)] = a \cdot C_2 \cdot \text{Cov}(R_i^o, c_i) > 0 \tag{6}$$

The total variance is minimized when:

$$\frac{d}{da}\text{Var}[\nabla_\theta J(\theta)] = 0 \tag{7}$$

This gives:

$$a^* = -\frac{C_2\text{Cov}(R_i^o, c_i)}{C_1/N} \tag{8}$$

Substituting $a^*$ back:

$$\text{Var}[\nabla_\theta J(\theta)] = \text{Var}[\nabla_\theta J_0(\theta)] - \frac{(C_2\text{Cov}(R_i^o, c_i))^2}{C_1/N} \tag{9}$$

$$= \text{Var}[\nabla_\theta J_0(\theta)]\left(1 - \frac{N(C_2\text{Cov}(R_i^o, c_i))^2}{C_1\text{Var}[\nabla_\theta J_0(\theta)]}\right)$$

Setting $\eta = \frac{N(C_2\text{Cov}(R_i^o, c_i))^2}{C_1\text{Var}[\nabla_\theta J_0(\theta)]} > 0$ completes the proof. $\square$

## C.2 THE PROOF OF PROPOSITION 2

Since $J$ is not necessarily concave, we use the following descent property. For sufficiently small $\alpha_t$:

$$J(\theta_{t+1}) \geq J(\theta_t) + \alpha_t \langle \nabla_\theta J(\theta_t), \hat{g}_t \rangle - \frac{L\alpha_t^2}{2}\|\hat{g}_t\|^2 \tag{10}$$

where $L$ is the Lipschitz constant of $\nabla_\theta J$.

Taking expectations, we may have:

$$\mathbb{E}[J(\theta_{t+1})] \geq \mathbb{E}[J(\theta_t)] + \alpha_t \mathbb{E}[\|\nabla_\theta J(\theta_t)\|^2] - \frac{L\alpha_t^2}{2}\mathbb{E}[\|\hat{g}_t\|^2] \tag{11}$$

Then the noise term can be bounded by:

$$\mathbb{E}[\|\hat{g}_t\|^2] = \mathbb{E}[\|\nabla_\theta J(\theta_t)\|^2] + \mathbb{E}[\|\hat{g}_t - \nabla_\theta J(\theta_t)\|^2] \leq G^2 + \sigma_g^2 \tag{12}$$

Rearranging and summing from $t = 1$ to $T$:

$$\sum_{t=1}^{T} \alpha_t \mathbb{E}[\|\nabla_\theta J(\theta_t)\|^2] \leq J(\theta^*) - \mathbb{E}[J(\theta_1)] + \frac{L}{2}\sum_{t=1}^{T} \alpha_t^2(G^2 + \sigma_g^2) \tag{13}$$

With $\alpha_t = \frac{\alpha}{\sqrt{t}}$, we have:

$$\sum_{t=1}^{T} \alpha_t = \alpha \sum_{t=1}^{T} \frac{1}{\sqrt{t}} \geq \alpha \int_1^T \frac{1}{\sqrt{x}}dx = 2\alpha(\sqrt{T} - 1) \tag{14}$$

$$\sum_{t=1}^{T} \alpha_t^2 = \alpha^2 \sum_{t=1}^{T} \frac{1}{t} \leq \alpha^2(1 + \ln T) \tag{15}$$

Equation 13 divided by $\sum_{t=1}^{T} \alpha_t$ and using the bounds above, we may have:

$$\mathbb{E}\left[\frac{1}{T}\sum_{t=1}^{T}\|\nabla_\theta J(\theta_t)\|^2\right] \leq \frac{2(J(\theta^*) - J(\theta_1)) + \alpha^2 G^2 + \alpha^2 \sigma_g^2}{\alpha\sqrt{T}} \tag{15}$$

Based on Proposition 1, we may have:

$$\sigma_g^2 \leq (1 - \eta)\sigma_{g,o}^2 \tag{16}$$

where $\sigma_{g,o}^2$ represents the variance of the stochastic gradient estimator for original GRPO objectives.

We finally have the following result:

$$\mathbb{E}\left[\frac{1}{T}\sum_{t=1}^{T}\|\nabla_\theta J(\theta_t)\|^2\right] \leq \frac{2(J(\theta^*) - J(\theta_1)) + \alpha^2 G^2 + \alpha^2 \sigma_g^2}{\alpha\sqrt{T}} \leq \mathbb{E}\left[\frac{1}{T}\sum_{t=1}^{T}\|\nabla_\theta J_o(\theta_t)\|^2\right] \tag{17}$$

$$\square$$

## D   IMPLEMENTATION DETAILS

The training is conducted on verl framework (Sheng et al., 2025) and the evaluation is conducted based on Lighteval toolkit (Habib et al., 2023). For RL training on Qwen2.5-7B and Llama3.1-8B and Mistral-7B-v0.1, $\alpha$ is set to 1, 0.5, and 0.5 while the learning rate is set to 1e-6, 2e-7, and 1e-7, respectively. Adam (Kingma & Ba, 2017) is adopted as optimizer. To avoid data contamination, it is ensured that the mathematical questions in the data for RL training training completely overlapped with those in the data for reward model training.

## E   COLD-START ISSUE

Table 3: Overall performance comparison on MATH-500, AIME2025, OlympiadBench, AMC-23, and GPQA with Llama3.1-8B. We report Pass@1 accuracy (%) and the average token length of response (#Tok). Bold denotes the best result.

| Datasets | MATH-500 | | AIME2025 | | OlympiadBench | | AMC-23 | | GPQA | | Average | |
|---|---|---|---|---|---|---|---|---|---|---|---|---|
| Metrics | Pass@1 | #Tok | Pass@1 | #Tok | Pass@1 | #Tok | Pass@1 | #Tok | Pass@1 | #Tok | Pass@1 | #Tok |
| SFT | 29.8 | 516.4 | 3.3 | 950.3 | 5.6 | 646.9 | 5.0 | 553.4 | 27.8 | 498.4 | 14.3 | 633.1 |
| GRPO | 12.6 | 2733.7 | **0.2** | 2513.6 | 3.4 | 3142.8 | 5.0 | 3412.3 | 25.8 | 2011.5 | 9.4 | 2762.8 |
| +Cos | 7.4 | 8.9 | 0.0 | 273.3 | 3.9 | 8.5 | 7.5 | 7.4 | **27.3** | 5.0 | 9.2 | 60.6 |
| +Kimi | 10.6 | 35.7 | 0.0 | 17.9 | **5.3** | 45.1 | **10.0** | 7.0 | 24.2 | 7.6 | 10.0 | 22.7 |
| +CRF | **13.2** | 89.6 | 0.0 | 136.5 | 4.6 | 123.9 | **10.0** | 106.4 | 26.8 | 80.2 | **10.9** | 107.3 |

Table 4: Overall performance comparison on MATH-500, AIME2025, OlympiadBench, AMC-23, and GPQA with Mistral-7B-v0.1. We report Pass@1 accuracy (%) and the average token length of response (#Tok). Bold denotes the best result.

| Datasets | MATH-500 | | AIME2025 | | OlympiadBench | | AMC-23 | | GPQA | | Average | |
|---|---|---|---|---|---|---|---|---|---|---|---|---|
| Metrics | Pass@1 | #Tok | Pass@1 | #Tok | Pass@1 | #Tok | Pass@1 | #Tok | Pass@1 | #Tok | Pass@1 | #Tok |
| SFT | 19.0 | 564.9 | 0.0 | 728.9 | 4.9 | 664.0 | 2.5 | 531.5 | 29.3 | 504.7 | 11.1 | 598.8 |
| GRPO | 7.6 | 490.6 | 0.0 | 1022.2 | 3.7 | 265.5 | 5.0 | 269.5 | 27.8 | 292.0 | 8.8 | 468.0 |
| +Cos | 4.0 | 16.2 | 0.0 | 8.2 | 1.6 | 209.4 | 2.5 | 6.3 | **31.8** | 6.2 | 8.0 | 49.3 |
| +Kimi | **10.6** | 39.3 | 0.0 | 7.7 | **4.0** | 16.0 | 5.0 | 10.9 | 30.8 | 10.0 | **10.1** | 16.8 |
| +CRF | 9.8 | 105.6 | 0.0 | 113.4 | 3.7 | 120.8 | **7.5** | 99.8 | 29.3 | 167.7 | **10.1** | 121.5 |

The training curves of Llama3.1-8B and Mistral-7B-v0.1 are depicted in Fig.6(a) and (b). We observe that on these two LLM backbones the reward of GRPO (i.e., Zero-R1-like training) is inferior to that of SFT+GRPO (i.e., R1-like training). This is probably because the base model struggles to follow specific formatting instructions (Liu et al., 2025b). Besides, as shown in Tab. 3 and 4, on Llama3.1-8B the R1-Zero-like training results in the generation of excessively verbose thinking processes. By contrast, our approach still demonstrates superior performance and higher token efficiency over the original GRPO and baseline methods.

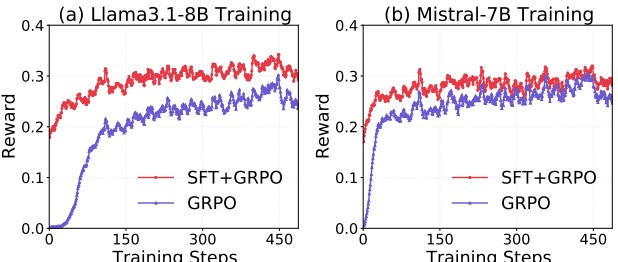

Figure 6: Training curves of GRPO (Zero-R1 like) v.s. SFT+GRPO (R1-like) on (a) Llama3.1-8B and (b) Mistral-7B-v0.1. The y-axis is outcome reward.

## F ETHICS STATEMENT

We acknowledge and adhere to the ICLR Code of Ethics.

## G REPRODUCIBILITY STATEMENT

First, to ensure reproducibility, the code and datasets for implementation are available on the anonymous link: https://anonymous.4open.science/r/CRM. Second, the dataset pre-processing step is detailed in section 3.2, Third, the experimental details are provided in Appendix D.

## H LLM USAGE

LLM is not used in any stage of the research and writing.

