# OpenReview forum: "Efficient Reasoning via Reward Model"
_ICLR.cc/2026/Conference — ICLR 2026 Conference Withdrawn Submission_

### Official Review · Reviewer_qKNM · 2025-10-27

**Soundness:** 2
**Presentation:** 2
**Contribution:** 1
**Rating:** 2
**Confidence:** 5

**Summary:**

This paper proposes a novel post-training pipeline for improving reasoning efficiency in LLMs by encouraging shorter reasoning paths. Specifically, the authors introduce a Conciseness Reward Model (CRM), a compact 3B model distilled from Qwen2.5-72B-Instruct, to evaluate the brevity of generated responses. The conciseness scores are then combined with difficulty scores through an annealing schedule to produce the final reward. By integrating CRM into the RL-based post-training stage, the resulting LLM is able to generate reasoning traces that are both accurate and concise.

**Strengths:**

1. Proposes a novel pipeline that incorporates a compact Conciseness Reward Model (CRM) to enable more efficient LLM reasoning.

2. Provides comprehensive empirical evaluations across multiple LLM backbones, including Qwen, LLaMA, and Mistral.

**Weaknesses:**

1. The proposed idea of distilling a compact CRM from a large and powerful LLM is intuitive but lacks rigorous analysis. It remains unclear how the large model reliably generates accurate conciseness scores for reasoning paths. Moreover, if training efficiency is not a concern, it would be valuable to justify why the authors do not directly use Qwen-72B-Instruct to compute conciseness scores.

2. The baseline comparison is incomplete. Several representative post-training variants, such as DAPO, should be included to better demonstrate how much accuracy is sacrificed when pursuing reasoning efficiency.

3. The reduction in the number of generated tokens achieved by CRM is relatively marginal compared to existing methods like COS and Kimi.

4. The reported accuracy of the proposed CRF model is notably low, especially on challenging mathematical reasoning benchmarks such as AMC23 and AIME25. For instance, a post-trained Qwen-7B-Base can typically achieve scores of 60+ on AMC23 and 12+ on AIME25, whereas the proposed CRF only reaches 55 and 6.7, respectively, indicating substantial performance degradation. It would also strengthen the paper to include results on the widely used AIME24 benchmark.

**Questions:**

Please see Weaknessnes.

---

### Official Review · Reviewer_gi4v · 2025-10-28

**Soundness:** 2
**Presentation:** 3
**Contribution:** 3
**Rating:** 4
**Confidence:** 3

**Summary:**

This paper addresses the problem of inefficient and overly verbose reasoning in large reasoning models (LRMs) trained with reinforcement learning. The authors observe two types of pathologies in prior length-penalized reward learning frameworks (e.g., RLVR, DAPO): **length collapse** (producing overly short, uninformative outputs) and **training collapse** (both reward and reasoning length drop, causing instability).

To mitigate these issues, the paper proposes an adaptive reward shaping framework based on a **Conciseness Reward Model (CRM)** and a **Conciseness Reward Function (CRF)**:

1. **Conciseness Reward Model (CRM):**
   A 3B-parameter model trained via supervised fine-tuning on 65K augmented mathematical reasoning samples to evaluate reasoning conciseness from 0.1–1.0.
2. **Conciseness Reward Function (CRF):**
   Combines the correctness reward \( R_i^o \) with the conciseness score \( c_i \), scaled by an annealing coefficient \( s \) and a difficulty factor \( d_q \):
   \[
   \hat{R}_i = R_i^o [1 + \alpha \, c_i (s + d_q)]
   \]
   Conciseness is only rewarded when the output is correct, preventing reward hacking.

The authors theoretically show that CRF reduces gradient variance and improves training stability, and empirically demonstrate strong gains in both accuracy and efficiency:
- +8.1% accuracy and −19.9% average token length compared with GRPO baseline on Qwen2.5-7B;
- consistent performance across Llama3-8B and Mistral-7B backbones;
- reduced reward variance and stabilized reasoning length distribution.

Overall, the paper presents a principled and empirically validated method for balancing correctness and conciseness in reasoning-focused reinforcement learning.

**Strengths:**

1. **Clear problem identification.**
   The paper explicitly defines two major failure modes in current reward-based reasoning optimization — *length collapse* and *training collapse*. These phenomena are real, empirically verified, and practically important for RLHF/RLVR research.

2. **Elegant reward formulation.**
   The proposed Conciseness Reward Function (CRF) only rewards brevity when the reasoning is correct, effectively preventing reward hacking and over-penalization of valid long reasoning.

3. **Balanced theoretical and empirical presentation.**
   Theoretical derivations (variance reduction and convergence proofs) support the empirical intuition. The paper maintains a good balance between formalism and practical validation.

4. **Strong practical motivation.**
   The method directly targets the computational inefficiency of chain-of-thought reasoning, making it relevant for scaling and deployment scenarios where token cost matters.

5. **Stable optimization behavior.**
   Empirical curves show smoother reward and length convergence than RLVR or DAPO, suggesting real improvement in RL training dynamics.

**Weaknesses:**

1. **Over-reliance on the Qwen family models.**
   All main experiments — including baselines, reward models, and CRM training — are built upon Qwen2.5 and Qwen-Math checkpoints.
   Since Qwen’s pre-training corpus contains ** similar reasoning data**, there is a strong possibility of **data contamination** or leakage.
   This compromises the fairness and generalizability of results, especially when all baselines share the same Qwen backbone.

2. **Limited model diversity.**
   Although Llama-3 and Mistral results are briefly mentioned, they are neither fully trained nor ablated.
   The claimed “cross-model generalization” is therefore weakly supported.

3. **Synthetic reward supervision.**
   The Conciseness Reward Model (CRM) is trained on pseudo-labels generated by another LLM (Qwen2.5), not human ratings.
   This introduces teacher bias — the CRM may inherit stylistic or reasoning biases from Qwen itself.

4. **Domain restriction.**
   All experiments are in **mathematical reasoning**.
   No tests are provided on code, logic, or general reasoning datasets, limiting the scope of applicability.

5. **Efficiency not fully quantified.**
   The work measures efficiency only in terms of *token length reduction*, not actual compute time or energy cost, which weakens the “efficient reasoning” claim.

6. **Potential reward circularity.**
   Since both correctness and conciseness signals originate from similar models (Qwen-RM and Qwen-CRM), the overall reward shaping pipeline lacks independence and may reinforce model-specific stylistic tendencies rather than genuine reasoning efficiency.

**Questions:**

1. **Model Dependence and Potential Data Leakage**
   Your experiments rely heavily on Qwen2.5, Qwen-Math, and Qwen-RM, which are known to contain large quantities of mathematical reasoning data (e.g., GSM8K, MathBench).
   How did you ensure that your evaluation datasets were not contaminated or overlapping with Qwen’s pretraining corpus?
   Would your conclusions hold if you repeated the experiments on models such as Llama-3 or Mistral that are less math-specialized?

---

2. **Independence of Reward Components**
   Both the correctness reward and the conciseness reward come from models derived from the Qwen family.
   Could this shared model lineage introduce circular bias into your training (i.e., the reward model reinforcing its own stylistic tendencies)?
   Have you tried using a different family of reward models to confirm independence?

---

3. **Conciseness Reward Model (CRM) Supervision Quality**
   The CRM is trained using pseudo-labels generated by Qwen2.5.
   How confident are you that these pseudo-labels reflect genuine human preferences for “conciseness” rather than model-specific stylistic bias?
   Did you consider human or cross-model calibration to improve robustness?

---

4. **Cross-domain and Cross-task Generalization**
   All reported results are in mathematical reasoning.
   Have you evaluated CoLD (or CRF) on other reasoning domains such as code generation, logic puzzles, or open-ended QA?
   If not, what limitations do you expect when transferring your method to such domains?

---

5. **Definition of Efficiency**
   You primarily report token-length reduction as evidence of reasoning efficiency.
   Have you measured real computational efficiency — e.g., wall-clock time, FLOPs, or energy savings — to validate that CRF truly improves inference efficiency rather than just shortening outputs?

---

6. **Training Stability and Hyperparameter Sensitivity**
   The annealing and difficulty coefficients (\( s \), \( d_q \), \( \alpha \)) appear central to stability.
   How sensitive is the method to these parameters?
   Would the model still converge stably if they were mis-specified or fixed constants?

Can you show the actual correlation between correctness and conciseness scores in your dataset?
How were baseline hyperparameters chosen? Can you show that other settings don't work better?
What's the wall-clock overhead of calling CRM during training?
Why only Pass@1? Most reasoning papers report Pass@k.
Would be useful to see human evaluation of reasoning quality and computational cost analysis of running CRM.

---

### Official Review · Reviewer_CLdK · 2025-10-30

**Soundness:** 2
**Presentation:** 3
**Contribution:** 2
**Rating:** 2
**Confidence:** 4

**Summary:**

This paper proposes a Conciseness Reward Model (CRM) and a Conciseness Reward Function (CRF) to improve reasoning efficiency in large reasoning models. By combining correctness-based rewards with learned conciseness signals, the method reduces redundant reasoning steps without hurting accuracy. Experiments on multiple reasoning benchmarks show that CRF achieves shorter responses and higher accuracy than prior efficient reasoning methods, effectively avoiding length and training collapse.

**Strengths:**

1. The paper is easy to follow and well structured
2. The proposed and trained reward model provide a new way for concise rewarding.

**Weaknesses:**

My main concerns lie in the **evaluation design**.

1. The primary results focus on *Pass@K*, which is not a standard metric for evaluating current Large Reasoning Models (LRMs). While Pass@K is meaningful, a paper aiming to balance **accuracy and reasoning efficiency** should also report *Average@K* for fairer comparison with prior work.
2. The evaluated models are not representative of today’s **reasoning or thinking models**. The reported token reduction (from ~500 to ~300) appears modest, especially considering that modern inference engines (e.g., VLLM, SGLang) can offset such differences with optimization.
3. The discussion section is insightful, but the connection between those analyses and the proposed **Conciseness Reward Model (CRM)** or **Conciseness Reward Function (CRF)** remains unclear. It would be helpful to clarify how these findings directly relate to the proposed methods.

**Questions:**

See weaknesses

---

### Official Review · Reviewer_9SfF · 2025-10-31

**Soundness:** 3
**Presentation:** 3
**Contribution:** 2
**Rating:** 2
**Confidence:** 4

**Summary:**

The paper proposes Conciseness Reward, a new reward view to encourage concise reasoning via a learned reward model.
The author trained a model (CRM) to score the conciseness of an answer, and incorporated this reward into the RL algorithm's reward based on difficulty and training stage, and validated the effectiveness of the method using GRPO.
The author conducted sufficient ablation experiments around the proposed method.

But the main experiment for verifying its effectiveness seems to have some issues, see Weakness and Question.
This method introduces a reward model; it may cause new reward hacking, such as skipping steps that should not be skipped in the inference process.
I do not recommend accepting this paper unless the author fully explains the effectiveness of their methods.

**Strengths:**

1. The paper proposes a novel perspective: optimizing with conciseness rewards.
2. The paper is well written, with clear motivation and mathematical presentation.
3. The experiments and ablation setting are reasonable.

**Weaknesses:**

The method is reasonable and simple, so my main question is about the experimental results.
The issue of the reproduced baseline raises doubts about the effectiveness of the proposed method.

The reproduced baseline in Table 2 differs significantly from the cited paper, especially the paper proposed Cos conducted same experiments on Llama-3.1 8B. The number of tokens shows that the baseline reproduced by the author has a length collapse, while the original Cos paper did not. This raises doubts about the correctness of the baseline experiment of Qwen-2.5.

**Questions:**

The method is simple, and the authors provide ablation experiments for reference. The main issue arises from the effectiveness of the method itself. Please refer to Weakness.

1. How would the training behavior and results be if a length penalty were used in place of the score of CRM?

---

### Note · Authors · 2025-11-12

I have read and agree with the venue's withdrawal policy on behalf of myself and my co-authors.